# Optimum Nitrogen, Phosphorous, and Potassium Fertilizer Application Increased Chrysanthemum Growth and Quality by Reinforcing the Soil Microbial Community and Nutrient Cycling Function

**DOI:** 10.3390/plants12234062

**Published:** 2023-12-03

**Authors:** Xinyan Fang, Yanrong Yang, Zhiguo Zhao, Yang Zhou, Yuan Liao, Zhiyong Guan, Sumei Chen, Weimin Fang, Fadi Chen, Shuang Zhao

**Affiliations:** 1College of Horticulture, Nanjing Agricultural University, Nanjing 210095, China2017202002@njau.edu.cn (Y.Z.);; 2Key Laboratory of Landscaping, Ministry of Agriculture and Rural Affairs, Nanjing 210095, China; 3Zhongshan Biological Breeding Laboratory, No. 50 Zhongling Street, Nanjing 210014, China; 4Tianjin Cement Industry Design and Research Institute Co., Ltd., Sinoma Technology and Equipment Group Co., Ltd., Tianjin 300400, China

**Keywords:** recommended NPK dose, nutrient uptake, soil nutrient properties, rhizosphere soil, diversity and function of soil microbiome

## Abstract

Nitrogen (N), phosphorus (P), and potassium (K) are three macronutrients that are crucial in plant growth and development. Deficiency or excess of any or all directly decreases crop yield and quality. There is increasing awareness of the importance of rhizosphere microorganisms in plant growth, nutrient transportation, and nutrient uptake. Little is known about the influence of N, P, and K as nutrients for the optimal production of *Chrysanthemum morifolium*. In this study, a field experiment was performed to investigate the effects of N, P, and K on the growth, nutrient use efficiency, microbial diversity, and composition of *C. morifolium*. Significant relationships were evident between N application rates, *C. morifolium* nutrient use, and plant growth. The N distribution in plant locations decreased in the order of leaf > stem > root; the distributions were closely related to rates of N application. Total P fluctuated slightly during growth. No significant differences were found between total P in the roots, stems, and leaves of *C. morifolium* vegetative organs. Principle component analysis revealed that combinations of N, P, and K influenced soil nutrient properties through their indirect impact on operational taxonomic units, Shannon index, and abundance of predominant bacterial taxa. Treatment with N, P, and K (600, 120, and 80 mg·plant^−1^, respectively) significantly improved plant growth and quality and contributed to the bacterial richness and diversity more than other concentrations of N, P, and K. At the flowering time, the plant height, leaf fresh weight, root dry weight, stem and leaf dry weight were increased 10.6%, 19.0%, 40.4%, 27% and 34.0%, respectively, when compared to the CK. The optimal concentrations of N, P, and K had a positive indirect influence on the available soil nutrient content and efficiency of nutrient use by plants by increasing the abundance of Proteobacteria, decreasing the abundance of Actinobacteria, and enhancing the potential functions of nitrogen metabolism pathways. N, P, and K fertilization concentrations of 600, 120, and 80 mg·plant^−1^ were optimal for *C. morifolium* cultivation, which could change environmental niches and drive the evolution of the soil microbial community and diversity. Shifts in the composition of soil microbes and functional metabolism pathways, such as ABC transporters, nitrogen metabolism, porphyrin, and the metabolism of chlorophyll II, glyoxylate, and dicarboxylate, greatly affected soil nutrient cycling, with potential feedback on *C. morifolium* nutrient use efficiency and growth. These results provide new insights into the efficient cultivation and management of *C. morifolium*.

## 1. Introduction

Chrysanthemum (*Chrysanthemum morifolium* Ramat.) originated in China and has become one of the four most ornamentally and economically important cut flowers globally [1]. With improved living standards, consumer demand for both cut flowers and potted plants has continued to increase. However, nutrients in the soil used to cultivate *C. morifolium* usually decline with ensuing cropping years [2], which results in declines in chrysanthemum growth and quality. The use of fertilizers that contain nitrogen (N), phosphorous (P), and potassium (K) has dramatically increased in the past century [3].

N, P, and K are three essential macronutrients and the most limiting factors in plant production. N affects the amino acid composition of proteins, which determines the rate of photosynthesis and, in turn, nutritional quality [4]. N-containing fertilizer is crucial for the nutrient balance of plants, which is essential for maximizing crop growth and yield. The application of commercial fertilizers containing N has been directly linked to crop yield and quality traits. The cost of N often accounts for a large proportion of the total agricultural input [5]. P is an important element that limits physiological metabolism and reproduction during growth. In general, the supply of N increases P uptake by many plants [6,7]. P uptake, rather than N uptake, in many different crops is the main source of variation in the N:P ratio [8]. K is mainly absorbed from the soil [9] and is important in plant physiological functions [10]. Mounting evidence points to the effects of optimum fertilization on plants, including plant height; tissue concentrations of N, P, and K in tropical trees [11]; wheat yield [12]; chrysanthemum production time [13]; rice nutrient use efficiency [14]; and soil nutrient status of flax (*Linum usitattisimum* L.) [15]. However, a field investigation reported that nearly 550 to 600 kg N·ha^−1^·yr^−1^ was applied to chrysanthemum plantations across the primary chrysanthemum-producing areas of China to allow for continuous growth during consecutive cropping years. The overapplication of N fertilizers is a major threat to agricultural production and ecosystem sustainability in plant production, as the excessive use of N fertilizers results in serious harm to the environment and soil health [16].

The soil microbiome plays a major role in maintaining soil health. Soil microorganisms are vital constituents of the plant rhizosphere, with a fundamental role in soil nutrient cycling and plant growth [17,18]. Cropping practices and fertilizer application can influence the nutrient availability of plants and alter soil community composition and biodiversity [19,20]. It is generally acknowledged that the growth and nutrition responses of plants to whole rhizosphere microbial communities are strongly associated with fertilization balance [21]. Long-term N application causes soil acidification, which may inhibit microbial growth and activity, and thus lead to the release of microorganism-derived phosphatases. In contrast, a reduced N level can significantly enhance bacterial growth and, potentially, plant quality compared to soils that receive abundant N [22,23]. Therefore, a better understanding of plant nutrition and nutrient interactions is critical for developing the best management practices for the high-yield production of chrysanthemum plants. Many studies have reported that different N sources or amounts affect plant biomass and soil nutrient status. However, few studies have addressed the effects of N, P, and K fertilization on chrysanthemum growth and quality, and the interaction with soil microbiota.

In this study, we systematically investigated chrysanthemum quality and the soil microbiome in response to different rates of N, P, and K application. The objectives of this study were to (1) examine the effects of N, P, and K nutrient combinations on chrysanthemum growth, nutrient absorption, and distribution at different growth stages; (2) determine the response of soil microorganisms to N, P, and K nutrient combinations; and (3) clarify the underlying relationships among N, P, and K nutrients; soil chemical attributes; chrysanthemum quality; and soil microorganisms. The results will provide a theoretical basis for improving chrysanthemum production through a balanced application of N, P, and K and lead to a better understanding of the roles and functions of microbes in the balanced fertilization of the soil ecosystems of the chrysanthemum rhizosphere.

## 2. Materials and Methods

### 2.1. Field Experimental Sites’ Description

This study was performed at the Chrysanthemum Germplasm Resource Preservation Center in Nanjing City, Jiangsu Province, China. The longitude is 118.85, and the latitude is 31.95. The clay soil had a pH of 5.56 and a specific conductance of 656.33 µS·cm^−1^, and it contained 27.1 g organic matter·kg^−1^, 44.33 mg available N·kg^−1^, 36.31 mg available P·kg^−1^ and 324.24 mg available K·kg^−1^.

### 2.2. Materials and Experimental Design

Before this experiment, we tested 4 different combinations of N, P, and K, resulting in a total of 45 treatments (5 N levels, 3 P levels, and 3 K levels). Based on the results of the preliminary orthogonal test of these 45 treatments, 5 representative treatments with significant differences were selected in this study. Prior to the main experiment, field experiments were performed during the growing seasons from April to October in 2021 and 2022. Before being transplanted to the field, seedlings of the standard cut chrysanthemum cultivar ‘Qinhuai Yulian’ were obtained by culturing in perlite at a spacing of 20 cm for 3 weeks in a greenhouse under a 16 h photoperiod and 70% relative humidity. The day and night temperatures were maintained at 28 °C and 22 °C, respectively. We used a randomized design with five N levels. Three replicates were used for each treatment. The five treatments are listed in Table 1.

After being transplanted in the field, ‘Qinhuai Yulian’ seedlings were fertilized every 15 days until the end of October. The fertilization treatment was divided into three periods: the seedling stage from 11 July to 11 August, the reproductive stage from 11 August to 30 September, and the flowering stage from 1 October to 30 October. The ratios of N, P, and K (hereafter NPK ratios) of the different treatments remained constant during each period. For all fertilization treatments, N-1, N-2, and N-3 represent the sampling stages at the seedling stage (30 d), reproductive stage (60 d), and flowering stage (90 d). 

### 2.3. Soil and Plant Sampling and Laboratory Analyses

Forty-five soil samples 0–20 cm in depth were collected from five treatments at 30 d, 60 d, and 90 d after planting. The rhizosphere soil from each plant was carefully collected by softly shaking with hands. A 15-plot experiment was set following a randomized complete block design with three replicates and five treatments. Composite samples were collected for nutrient analyses. Ammonium N was determined by performing KCL extraction–indigo blue colorimetry on an Alpkem Autoanalyzer (OI Analytics, College Station, TX, USA). Available P was determined using the sodium bicarbonate extraction–molybdenum antimony coloration–spectrophotometric method. Available K was determined using the ammonium acetate extraction–flame photometric method. Soil that was not tightly adhered to the roots was shaken off. The remaining soil was collected, sieved, and stored at −80 °C for genomic DNA extraction. 

For each growth stage, six uniform plants from each plot were sampled. Shoot height, shoot diameter, shoot dry weight, root dry weight, leaf width and length, flower diameter, and ray floret number were measured. The nutrient content of plants was analyzed. Total N concentration was determined using a model AA3 continuous flow analyzer (SEAL Analytical, Norderstedt, Germany). Total P and K concentrations were analyzed using an Optima 8000 plasma emission spectrometer (PerkinElmer, Waltham, MA, USA). Nutrient-use efficiency was calculated.

### 2.4. DNA Extraction, PCR Amplification, and Illumina Sequencing

Genomic DNA was extracted from 250 mg rhizosphere soil samples using a Power Soil DNA Isolation Kit (MoBio Laboratories, Carlsbad, CA, USA). The concentration and integrity of the resulting DNA were determined using a NanoDrop 2000 ultraviolet spectrometer (Thermo Fisher Scientific, Waltham, MA, USA). The V3+V4 region of the bacterial 16S rRNA gene was amplified using the gene-specific primers 341F (5′-CCTACGGGNGGCWGCAG-3′) and 806R 134 (5′-GGACTACHVGGGTATCTAAT-3′). The internal transcribed spacer (ITS)2 region of the fungal DNA was targeted by the primers KYO2F (5′-GATGAAGAACGYAGYRAA-3′) and ITS4R (5′-TCCTCCGCTTATTGATATGC-3′). Purified amplicons were pooled at equimolar concentrations and subjected to paired-end sequencing (2 × 250) on an Illumina platform, according to standard protocols. All reactions were performed in a total volume of 50 μL, containing 10 μL of 5× Q5@ Reaction Buffer, 10 μL of 5× Q5@ High GC Enhancer, 1.5 μL of 2.5 mM dNTPs, 1.5 μL of each primer (10 μM), 0.2 μL of Q5@ High-Fidelity DNA Polymerase, and 50 ng of template DNA, raised to 50 μL. Thermal cycling conditions were as follows: 95 °C for 5 min, followed by 30 cycles at 95 °C for 1 min, 57 °C for 1 min, and 72 °C for 1 min, and final extension at 72 °C for 7 min. All library preparations were performed using an Illumina HiSeq2500 platform at Genedenovo Biotechnology Co., Ltd. (Guangzhou, China). 

### 2.5. Statistical Analyses

The necessary computations were performed using routines implemented in Microsoft Excel 2017. Statistical analyses of all parameters were performed using the IBM SPSS statistical software package version 20 (IBM Corporation, Armonk, NY, USA). Data from each treatment were analyzed using a one-way analysis of variance (ANOVA). Duncan’s multiple-range test was used to assign significance to differences (*p* < 0.05) between treatment means. Raw reads were filtered using FASTP. Paired-end clean reads were merged as raw tags using FLSAH (version 1.2.11), and noisy sequences were filtered using the QIIME (version 1.9.1) pipeline to obtain high-quality clean tags. The clean tags were searched against a reference database to perform reference-based chimera verification using the UCHIME algorithm. All chimeric tags were removed, and the final effective tags were obtained, which were clustered into operational taxonomic units (OTUs) of ≥97% similarity using the UPARSE pipeline. A representative sequence from each OTU was selected, and Ribosomal Database Project (RDP) classifiers (RDP Bacterial 16S database for 16S rRNA data and UNITE Fungal ITS database for ITS data) were used for taxonomic characterization. Principal component analysis (PCA) was performed in R using the Vegan package (version 3.0.2). Differences in the bacterial and fungal communities between treatments were tested using an analysis of similarities (ANOSIM). The Kyoto Encyclopedia of Genes and Genomes (KEGG) pathway analysis of bacterial OTUs was performed using Tax4Fun (version 1.0), and fungal OTUs were inferred using FUNGuild. 

## 3. Results

### 3.1. Effects of Different NPK Ratios on Growth and Quality of Chrysanthemum Plants

The application of N, P, and K positively influenced the growth and quality of chrysanthemum plants (Table 2). In the seedling stage (30 d), the greatest chrysanthemum growth indices of plant height, stem fresh/dry weight, and leaf fresh/dry weight were observed in the N_3_P_2_K_1_ treatment, with the root fresh weight and dry weight increasing by 38.88% and 53.76%, compared with CK_1_. At the reproductive stage (60 d), a higher chrysanthemum growth index was observed for N_3_P_2_K_1_. At the flowering stage (90 d), the greatest plant height (101.27 cm) was detected using N_2_P_1_K_2_, whereas the greatest leaf fresh weight and root dry weight were observed using N_3_P_2_K_1_ (Table 2). 

The application of different NPK ratios significantly affected chrysanthemum flower quality (Table 3). The highest chrysanthemum flower fresh weight, flower dry weight, flower diameter, flower height, and maximum leaf area were obtained using N_3_P_2_K_1_, followed by N_2_P_1_K_2_ and N_1_P_1_K_1_. The flower diameter in the treatment N_3_P_2_K_1_ was 131.73 mm, which was significantly different from that of the other treatments. The longest flower stem length was recorded in the treatment N_3_P_2_K_1_, which was 1.20 times longer than that in CK.

### 3.2. Available Nutrient Content of Cultivation Soil

The results show that the application of different NPK ratios affected the ammonium N, available P, and available K contents in the chrysanthemum cultivation soil (Figure 1). In the seedling stage (30 d), the content of soil ammonium N displayed no significant differences among treatments, except for N_3_P_2_K_1_. In N_3_P_2_K_1_, ammonium N was lowest in the reproductive stage (60 d) and flowering stage (90 d). In both stages, the content of soil ammonium and nitrogen displayed a trend of an initial increase followed by a decrease (Figure 1a). The increase in the N_2_P_1_K_2_ treatment was 52.87% compared to CK. The highest content of available P was recorded for the CK and N_4_P_2_K_2_ treatments (82.36 mg/kg) in the reproductive stage (60 d) of the chrysanthemum (Figure 1b). The greatest available K was found for the N_2_P_1_K_2_ treatment (233.17 mg/kg) in the seedling stage (30 d). This increase was 1.17 times greater than that of the CK. No significant differences were evident with N_3_P_2_K_1_ and N_4_P_2_K_2_ (Figure 1c).

### 3.3. Effects of Different NPK Ratios on Chrysanthemum Nutrient Uptake and Tissue Distribution

Different NPK ratios had significant effects on total N, P, and K uptake and distribution in the roots, stems, leaves, and flowers of chrysanthemum plants during the entire growth period (Figure 2). At the seedling stage (30 d), the greatest total N content was obtained using N_3_P_2_K_1_. The total N content in the roots, stems, and leaves increased by 22.18%, 25.55%, and 21.1%, respectively, compared with CK (Figure 2A,a). In the reproductive stage (60 d), the highest total N content was obtained using N_3_P_2_K_1_. In this treatment, the total N content in the roots, stems, and leaves increased by 9.82%, 8.47%, and 13.34%, respectively, compared with CK. During the flowering stage (90 d), the N ratio increased the total N content in flowers by 64.3% using N_3_P_2_K_1_ (Figure 2a). N uptake was appreciably higher with the application of N, P, and K at the flowering stage. The total N content was highest in flowers using N_3_P_2_K_1_ (Figure 2D). Lower NPK ratios resulted in lower N uptake. The total P contents of the different treatments are shown in Figure 2B. In the seedling stage, the highest total P content was 2.13 mg·g^−1^ in roots and 2.99 mg·g^−1^ in leaves using N_4_P_2_K_2_, followed by N_2_P_1_K_2_. The respective P content was 8% and 21.5% higher, compared to the CK value (Figure 2B). In the reproductive stage, the total P content was highest for the N_3_P_2_K_1_ treatment (3.85 mg/g). P content was highest in the N_1_P_1_K_1_ treatment in the flowering stage, followed by N_2_P_1_K_2_. P content was lowest in the roots, stems, and leaves of plants treated with N_4_P_2_K_2_ at the flowering stage. There was no significant change in total P content during the later growth period (Figure 2b). The addition of P increased the total P content in flowers by 10.03%. The higher P uptake was the result of significantly higher dry matter production at 90 d (Table 2). The total K content in the roots, stems, and leaves fluctuated irregularly (Figure 2C). The distribution of K was ranked as leaf > stem > root (Figure 2c). The highest total K content in flowers was found in the N_1_P_1_K_1_ treatment (Figure 2C), which was 9.28% higher than that of CK.

### 3.4. Effects of NPK Ratios on Chrysanthemum Rhizosphere Soil Microbiota

#### 3.4.1. Shared and Unique OTUs

The application of different NPK ratios significantly increased the bacterial richness and diversity in the chrysanthemum rhizosphere. After further screening and clustering based on the 97% similarity threshold, a total of 4055 bacterial OTUs were observed in CK_3_, N1-3 (N_1_P_1_K_1)_, N2-3 (N_2_P_1_K_2_), N3-3 (N_3_P_2_K_1_), and N4-3 (N_4_P_2_K_2_) in the flowering stage (Figure 3). The 1504 shared OTUs (Figure 3a) accounted for 37.1% of the total OTUs. The N_3_P_2_K_1_ treatment displayed a higher number of unique OTUs *(n* = 284) than the other treatments. A total of 1215 fungal OTUs were observed in the treatments. The 337 shared OTUs accounted for 27.7% of the total observed fungal OTUs (Figure 3b). CK_3_ and N1-3 had more unique OTUs than the other treatments, indicating that the addition of NPK resulted in fewer fungal species than in CK.

#### 3.4.2. Microbial Diversity

The amendment of soil with fertilizer containing N, P, and K affected the bacterial and fungal richness and diversity in the chrysanthemum rhizosphere (Table 4). The alpha diversity index based on bacterial species richness varied among treatments (*p* < 0.05), especially between the CK- and NPK-treated rhizosphere soil samples. At the seedling stage (30 d), CK1 and N_1_P_1_K_1_ resulted in higher values for the bacterial Shannon–Weaver, Faith’s phylogenetic diversity, and Chao1 richness indices than the other treatments. At the reproductive stage (60 d), the application of N, P, and K contributed to a significant increase in bacterial alpha diversity indices (Shannon, Faith’s phylogenetic diversity, Chao1, and Evenness indices), with no significant differences in alpha diversity indices among the various NPK ratios. At the flowering stage (90 d), the highest Shannon, Faith’s phylogenetic diversity, and Chao1 indices were found for N_3_P_2_K_1_, which produced a more even bacterial population (0.79).

Fungal diversity varied considerably across the rhizosphere using the various NPK ratios. At the seedling stage (30 d), the lowest fungal alpha diversity indices were detected in N_4_P_2_K_2_, followed by N_3_P_2_K_1_. At the flowering stage (90 d), the N_3_P_2_K_1_ treatment resulted in the lowest values of the Shannon–Weaver, Faith’s phylogenetic diversity, and Chao1 richness indices, as well as the highest Evenness index (Table 4).

The beta diversity of microbial communities in chrysanthemum rhizosphere soils treated with different NPK ratio fertilization treatments was further analyzed. The principal coordinate analysis (PCoA) plot of microbial communities revealed a clear separation among the groups of chrysanthemum soils using various NPK ratios (Figure 4). The findings indicate differences in the microbial community structures of rhizosphere soils among the different fertilization treatments. For bacteria, compared with those at 60 d (N_X_-2) and 90 d (N_X_-3), the microbial structure among all treatments was similar at 30 d (N_X_-1) (Figure 4a). For fungi, the degree of aggregation of the sample points from each other was relatively low, and the communities of all treatments in each period were similar, indicating that the communities were greatly affected by fertilizer management (Figure 4b).

#### 3.4.3. Microbiome Composition

The overall taxonomic complexity of the chrysanthemum rhizosphere soil microbiome at the phylum level is shown in Figure 5. The 10 most abundant phyla of bacteria in descending order were Proteobacteria, Actinobacteria, Gemmatimonadetes, Planctomycetes, Firmicutes, Acidobacteria, Patescibacteria, Epsilonbacteria, Chloroflexi, and Bacteroidetes. These phyla accounted for more than 75% of the species predicted to be represented based on ribosomal gene sequencing (Figure 5a). Proteobacteria and Actinobacteria were dominant across all treatments investigated (Figure 5a). During the growing period of chrysanthemum, the abundance of Proteobacteria decreased gradually, whereas N_3_P_2_K_1_ treatment significantly (*p* < 0.05) increased the relative abundance of Proteobacteria and decreased the abundance of Actinobacteria compared to other treatments. The abundance of Proteobacteria and Actinobacteria in the chrysanthemum growth stages is ranked as seedling stage > reproductive growth stage > flowering stage. In addition, N_3_P_2_K_1_ treatment had the strongest effect on Epsilonobacteria abundance.

In comparison, the abundant fungal phyla in the chrysanthemum rhizosphere ranked in descending order as Ascomycota, Anthophyta, Ciliophora, Basidiomycota, Chlorophyta, and Mucoromycota, which comprised more than 90% of the species predicted to be represented based on ITS gene sequencing reads (Figure 5b). Ascomycota and Anthophyta were the dominant fungal phyla among all treatments. The N_2_P_1_K_2_ and N_4_P_2_K_2_ treatments favored the growth of Ascomycota, with maximum values of 86.01% and 82.01%, respectively. Throughout the growth stages, Ascomycota exhibited gradually increasing trends, whereas Anthophyta exhibited fluctuating trends.

#### 3.4.4. KEGG Pathway Function Prediction of Microbial Communities

Fertilization with N, P, and K significantly affected the functional gene abundance of bacteria and fungi in the chrysanthemum rhizosphere soil. Functions of the soil microbiome related to ABC transporters and the metabolism of N, porphyrin, chlorophyll II, glyoxylate, and dicarboxylate were increased in the reproductive and flowering stages of N_3_P_2_K_1_. Red clusters were functionally enriched in rhizosphere soils and were involved in the metabolism of cysteine and methionine; homologous recombination; and pyrimidine, ribosome, and peptidoglycan biosynthesis. The function of the soil microbiome related to glycine, serine, and threonine metabolism increased in the N_3_P_2_K_1_ treatment, which was significantly different from that of the other treatments (Figure 6a). Fungal analysis indicated that the number of pathways was significantly lower than that of bacterial pathways across all treatments. The functions of the soil fungal microbiome related to dung saprotroph–plant saprotroph–wood saprotroph, animal pathogen–soil saprotroph, and plant pathogen were highest in CK and N1 (Figure 6b).

#### 3.4.5. The Association between Microbial Communities and Soil Environmental Factors and Growth Indexes

Changes in the microbial community structure can be ascribed to variations in soil properties caused by the addition of N, P, and K. The results obtained in this study showed that plant height was the predominant factor explaining the variation in the bacterial community structure. Ammonium N and available P contents were also affected by the bacterial community structure (Figure 7a). Plant height was the predominant factor affecting the fungal community structure. Soil ammonium N content significantly affected the fungal community structure, indicating that the community structure was greatly affected by N fertilizer management (Figure 7b). Slight differences were evident between the root fresh weight and root dry weight, root total N, stem total N, and leaf total N in bacterial and fungal redundancy analyses.

## 4. Discussion

N, P, and K are limiting factors for plant growth and crop production. Low fertilizer application rates have been noted as the main cause of depleted soil nutrients and relatively stagnant crop yields [24]. However, excessive fertilizer inputs in the planting process induce rapid growth and high-yield crops, resulting in low fertilizer efficiency [25]. Our results indicate that the highest values of the height and dry biomass of Chrysanthemum ‘Qinhuai Yulian’ were obtained upon treatment with N3 (600 mg·plant^−1^), P2 (120 mg·plant^−1^), and K1 (300 mg·plant^−1^) among the tested treatments. The highest growth and flower quality indices were also detected in the N_3_P_2_K_1_ treatment in the flowering stage. The plant height, root fresh weight, and dry weight were lower in the NPK ratio with the overapplication of N_4_P_2_K_2_. The findings indicate that N_3_P_2_K_1_ is the optimal fertilizer treatment for the growth and quality of ‘Qinhuai Yulian’ chrysanthemum. Similar observations of increased growth indices as a result of balanced fertilization with N, P, and K have been reported in maize [26] and peanut [27].

N affects the amino acid composition of proteins and, in turn, their nutritional uptake and quality [28]. The results obtained in this study show that the content of soil ammonium N significantly fluctuated with different N application rates, with the greatest index evident for ammonium N exposed to N_2_P_1_K_2_ in the seedling stage, and for the reproductivity of N_3_P_2_K_1_. These results can be attributed to the availability of N, which enhances N uptake by plants to satisfy their ammonium N needs during different developmental stages [29]. With plant growth, the total N content in the roots, stems, and leaves decreased gradually, with the distribution ranking as leaf > stem > root. The data are similar to those reported by Hocking et al. [30] in Linola (low-linolenic acid linseed) and Ea Akinrinde et al. [31] in pineapple. In addition, our results show that the optimal application of fertilizer containing N_3_P_2_K_1_ had a clear advantage over other NPK ratios in N uptake, indicating that N accumulation increased with N concentration [32]. Total P fluctuated slightly throughout the growth. The highest index of the content of available P in the seedling stage was recorded for N_4_P_2_K_2_, and was lowest in the flowering stage for the N_4_P_2_K_2_ treatment, indicating that supplementation with a high concentration of P fertilizer is better for high P efficiency in the earlier stages of reproductive development, while a decline in the efficiency of P use at higher P concentrations in the flowering stages can be attributed to the low uptake of applied P by ‘Qinhuai Yulian’ chrysanthemum [33]. The total K content in the roots, stems, and leaves of chrysanthemum fluctuated irregularly. With chrysanthemum growth, the total K was transferred from nutritional organs to reproductive organs. The available K was highest using the N_2_P_1_K_2_ treatment in the seedling stage and was gradually reduced in the stems and leaves, indicating that the total K was transferred to the flower to promote the formation of the chrysanthemum plant [34,35].

The soil microbial activity is an important indicator of soil quality and health [36] and is sensitive to environmental changes, such as tillage [37], fertilization [38], seasonal variation [39], and plant type [40]. The application of N, P, and K provides excess N to soil microorganisms, possibly affecting soil microbial abundance [19]. Our results show that, during all growth stages, the diversity of bacteria in the chrysanthemum rhizosphere soil exhibited gradually increasing trends, while fungi exhibited gradually decreasing trends, indicating that differences in the NPK ratios have a crucial and direct impact on the diversity and abundance of microbiota in the rhizosphere soil [2]. The observation that N_3_P_2_K_1_ resulted in more bacterial OTUs and increased bacterial diversity suggests that the use of the optimal NPK ratio could maintain sustainable microbial soils in the chrysanthemum rhizosphere, leading to a more stable system [41,42]. Moreover, the present results also show that Proteobacteria and Actinobacteria were the two most abundant bacterial phyla across all the treatments investigated [43,44]. N_3_P_2_K_1_ significantly (*p* < 0.05) increased the relative abundances of Proteobacteria and decreased the abundance of Actinobacteria [45,46], indicating that the optimal NPK ratio promotes the abundance of Proteobacteria, which increases N cycling in the rhizosphere soil. Actinobacteria residing in soil produce active compounds and synthesize a wide range of bioactive compounds, including antibiotics, in the rhizosphere. Many of these compounds are important in agriculture. The use of an optimal NPK ratio maintains a more stable rhizosphere soil system for chrysanthemum plants. This function is negatively affected by the decreasing abundance of actinobacteria.

A correlation analysis performed in the present study revealed that the relative abundances of Proteobacteria in the chrysanthemum rhizosphere were positively correlated with soil ammonium N (Pearson’s *p* < 0.01, Figure 8a); the abundance of Actinobacteria was positively correlated with available P (Pearson’s *p* < 0.01); and the abundance of Chloroflexi was positively correlated with the plant height, root fresh weight, and root dry weight [47]. For fungi, the relative abundances of Ciliophora and Chlorophyta were positively correlated with soil ammonium N (Pearson’s *p* < 0.01, Figure 8b). The relative abundance of Ascomycota was positively correlated with the plant height, root fresh weight, and root dry weight. These data may indicate more copiotrophic conditions in the late-stage compared to early-stage chrysanthemum soils. Our results indicate that soil ammonium N and root biomass may promote the relative abundance of bacteria and fungi in the chrysanthemum rhizosphere. ABC transporters and the metabolic pathways of N, porphyrin, chlorophyll II, glyoxylate, and dicarboxylate observed with N_3_P_2_K_1_ treatment were predicted to have potential functions in nitrogen cycling and plant growth [48]. Therefore, it is necessary to further study the key functional genes involved in N cycling [49,50]. These results coincide with the trend of chrysanthemum production, further suggesting a close relationship between chrysanthemum production and soil chemical and biological measurements [51], and they can provide a better understanding of the importance of using the optimum NPK ratio to promote the soil microbial diversity, thereby increasing chrysanthemum production.

## 5. Conclusions

Growth indices, flower quality, soil fertility, and nutrient uptake of ‘Qinhuai Yulian’ chrysanthemum were significantly promoted using an optimal NPK ratio during growth stages. The N_3_P_2_K_1_ (N:P:K = 600:120:300 mg·plant^−1^) ratio significantly improved chrysanthemum productivity and had a greater effect on the rhizosphere soil’s bacterial and fungal community structure. Compared to the other treatments, N_3_P_2_K_1_ had greater impacts on the soil microbial abundance, diversity, composition, and N metabolism. The application of N, P, and K is considered a key factor in controlling soil fertility, plant production, and microbial diversity.

## Figures and Tables

**Figure 1 plants-12-04062-f001:**
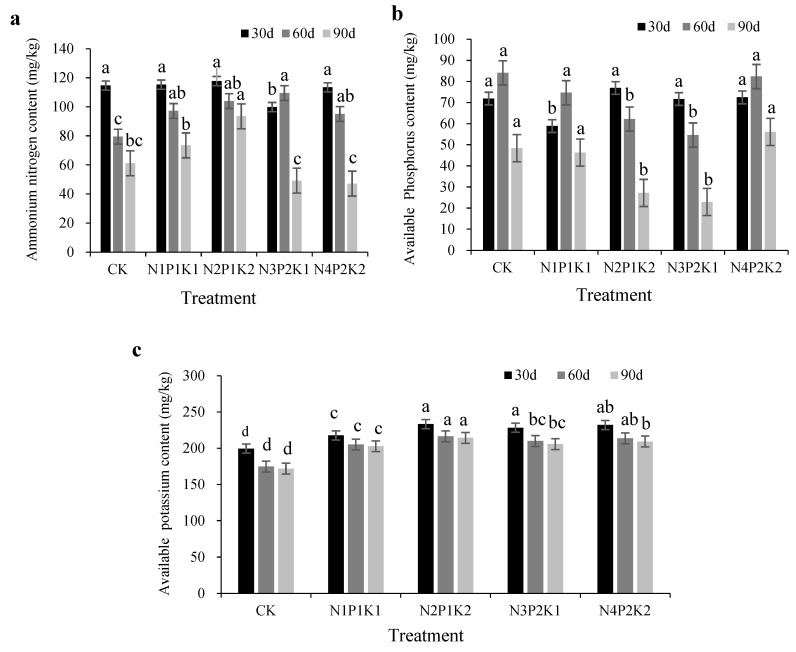
Effect of different NPK ratios on the available nutrient content of chrysanthemum cultivation soil. Note: (**a**) is the content of ammonium nitrogen (N), (**b**) is the content of available phosphorus (P); (**c**) is the content of available potassium (K). Different lowercase letters on bar chart indicate significant differences among the treatments (*p* < 0.05).

**Figure 2 plants-12-04062-f002:**
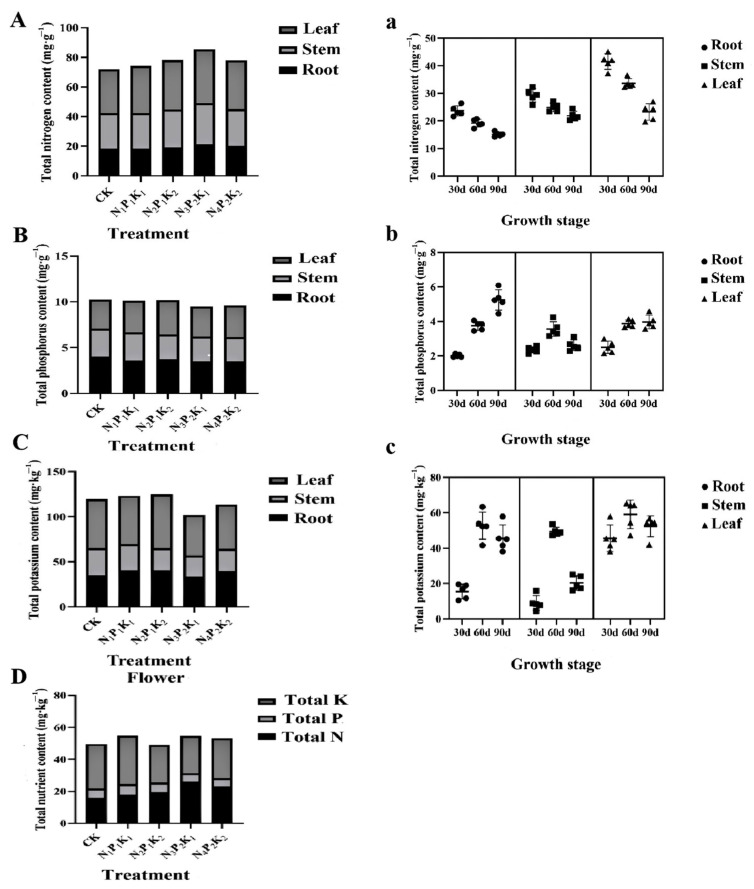
Effects of different NPK ratios on the uptake and tissue distribution of N, P, and K in chrysanthemum plants: (**A**,**a**) total nitrogen (N) content and N distribution in different organs and growth stages; (**B**,**b**) total phosphorus (P) content and P distribution in different organs and growth stages; (**C**,**c**) total potassium (K) content and K distribution in different organs and growth stages; (**D**) total contents of N, P, and K in flowers. (**A**–**D**) correspond to differences of total nitrogen in root + stem + leaf, total phosphorus, and total potassium in root + stem + leaf between treatments at the flowering stage; (**a**–**c**) correspond to differences of total nitrogen, total phosphorus, and total potassium content between seedling (30 d), reproductive (60 d) and flowering stages (90 d) in the root, stem, and leaf, respectively.

**Figure 3 plants-12-04062-f003:**
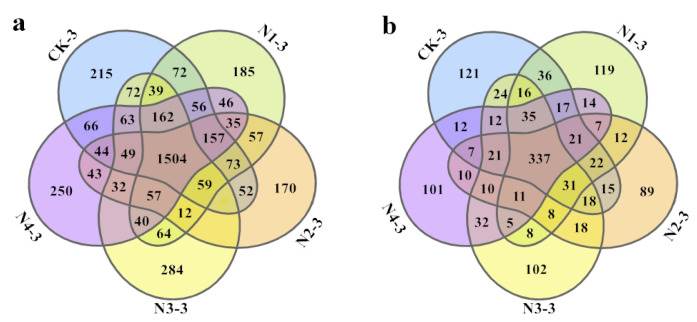
Venn diagrams showed shared unique OTUs of (**a**) bacteria and (**b**) fungi in the soils after treatments with different NPK ratios in the flowering stage (90 d). N_X_-1, N_X_-2, and N_X_-3 represent sampling at the seedling stage (30 d), reproductive stage (60 d), and flowering stage (90 d), respectively.

**Figure 4 plants-12-04062-f004:**
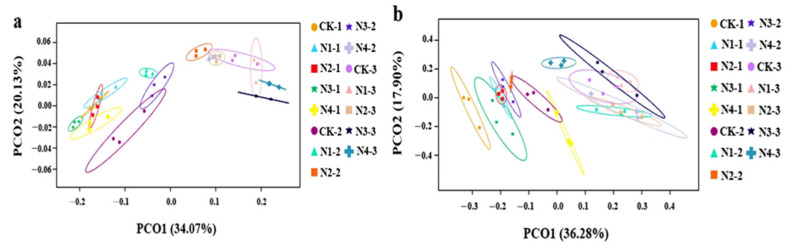
Principal coordinates analysis (PCoA) plot of (**a**) bacterial and (**a**) fungal community structures based on Bray–Curtis differences.

**Figure 5 plants-12-04062-f005:**
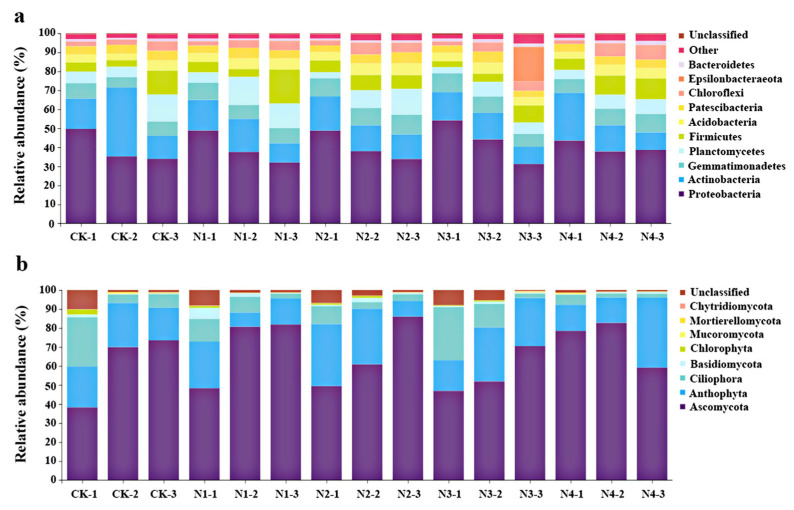
Relative abundance of (**a**) bacterial and (**b**) fungal phyla in the chrysanthemum rhizosphere using different NPK fertilization ratios.

**Figure 6 plants-12-04062-f006:**
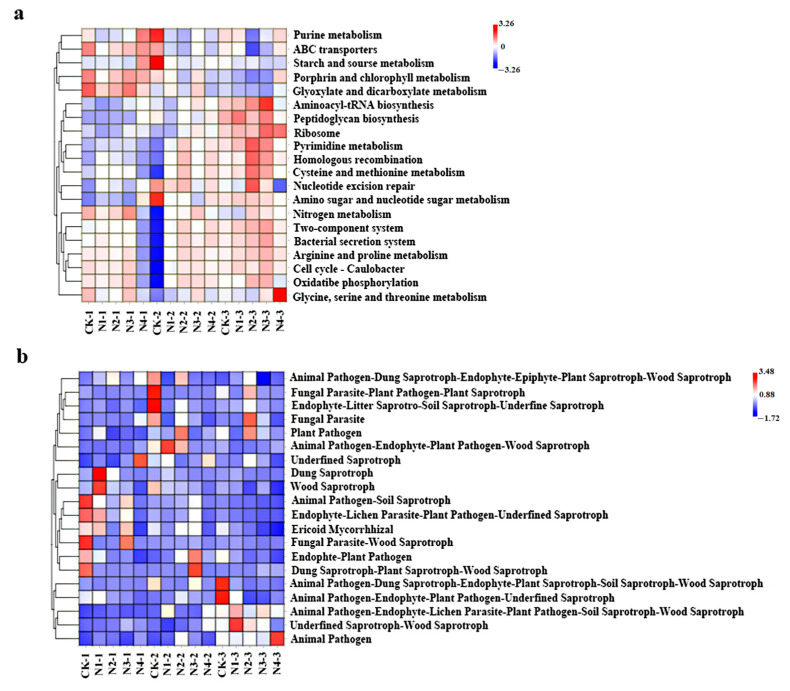
Heatmap of bacterial (**a**) and fungal (**b**) distribution of the top-20 most abundant gene-predicted functional profiles in the KEGG pathway present in the chrysanthemum rhizosphere soil using various NPK ratios. Red and green represent high and low enrichment of function abundance, respectively.

**Figure 7 plants-12-04062-f007:**
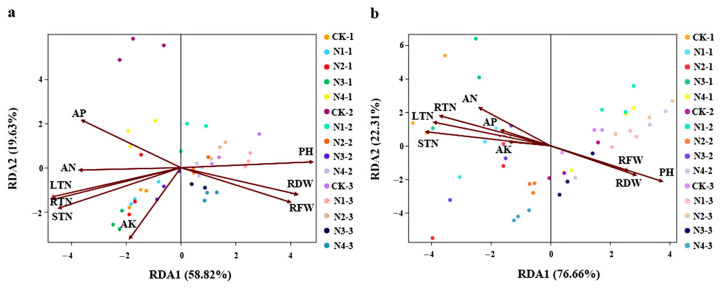
Redundancy analysis (RDA) demonstrating the relationships between soil environmental factors, growth indexes, and (**a**) soil bacterial and (**b**) fungal community structures among soil samples. Note: PH, RFW, and RDW indicate plant height, root fresh weight, and root dry weight, respectively. AN, AP, and AK indicate soil ammonium nitrogen, available phosphorus, and potassium, respectively. RTN, STN, and LTN indicate the total nitrogen content of roots, stems, and leaves, respectively.

**Figure 8 plants-12-04062-f008:**
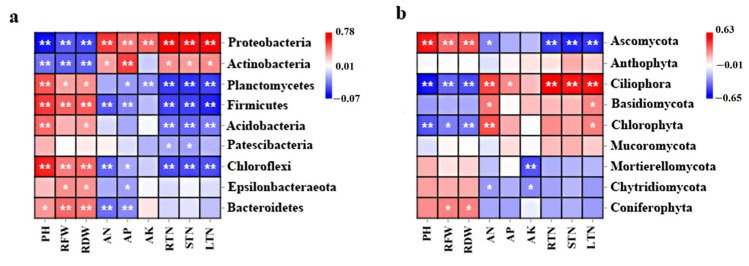
Correlation coefficients between soil environmental factors, growth indices, and relative abundances of bacterial (**a**) and fungal (**b**) communities in chrysanthemum rhizosphere. Notes: Significant correlation coefficients are indicated * *p* < 0.05 and ** *p* < 0.01 based on Pearson’s correlation test. PH, RFW, and RDW indicate plant height, root fresh weight, and root dry weight, respectively. AN, AP, and AK indicate soil ammonium nitrogen, available phosphorus, and potassium, respectively. RTN, STN, and LTN indicate the total nitrogen content of the roots, stems, and leaves, respectively.

**Table 1 plants-12-04062-t001:** NPK treatment design in this study.

Treatment	N (Urea, 46% N) mg·plant^−1^	P (18% P_2_O_5_) mg·plant^−1^	K (50% K_2_O) mg·plant^−1^
Control Check (CK)	0	0	0
N_1_P_1_K_1_	200	80	300
N_2_P_1_K_2_	400	80	500
N_3_P_2_K_1_	600	120	300
N_4_P_2_K_2_	800	120	500

**Table 2 plants-12-04062-t002:** Effect of different NPK ratios on the growth index of ‘Qinhuai Yulian’.

Stage and Treatment		Plant Height (cm)	Stem Diameter (mm)	Root Fresh Weight (g)	Stem Fresh Weight (g)	Leaf Fresh Weight (g)	Root Dry Weight (g)	Steam Dry Weight (g)	Leaf Dry Weight (g)
Seedling stage (30 d)	CK_1_	14.89 ± 0.46 c	2.49 ± 0.51 b	0.33 ± 0.16 b	1.15 ± 0.15 b	4.56 ± 0.50 b	0.043 ± 0.010 c	0.17 ± 0.024 b	0.45 ± 0.055 b
	N_1_P_1_K_1_	19.26 ± 0.28 b	3.16 ± 0.25 a	0.45 ± 0.11 b	2.40 ± 0.30 a	8.52 ± 1.24 a	0.077 ± 0.018 ab	0.28 ± 0.038 a	0.81 ± 0.14 a
	N_2_P_1_K_2_	20.60 ± 0.26 a	3.26 ± 0.41 a	0.38 ± 0.17 b	2.66 ± 0.51 a	9.68 ± 1.31 a	0.067 ± 0.016 b	0.34 ± 0.039 a	0.91 ± 0.12 a
	N_3_P_2_K_1_	20.99 ± 0.26 a	3.45 ± 0.20 a	0.54 ± 0.22 a	2.53 ± 0.64 a	9.52 ± 1.29 a	0.093 ± 0.018 a	0.34 ± 0.062 a	0.94 ± 0.17 a
	N_4_P_2_K_2_	19.67 ± 0.40 b	3.40 ± 0.45 a	0.41 ± 0.20 b	2.48 ± 0.61 a	8.71 ± 0.90 a	0.072 ± 0.020 b	0.33 ± 0.056 a	0.92 ± 0.13 a
Reproductive stage (60 d)	CK_2_	51.44 ± 1.66 c	3.76 ± 0.31 b	0.51 ± 0.26 b	8.96 ± 1.21 b	21.85 ± 3.64 d	0.15 ± 0.048 b	1.59 ± 0.33 b	2.17 ± 0.39 c
	N_1_P_1_K_1_	59.02 ± 1.42 b	4.32 ± 0.56 a	0.48 ± 0.078 b	10.69 ± 1.65 a	26.30 ± 1.18 bc	0.14 ± 0.023 b	1.76 ± 0.28 b	2.44 ± 0.45 bc
	N_2_P_1_K_2_	61.91 ± 1.91 a	4.32 ± 0.51 a	0.50 ± 0.11 b	11.94 ± 1.23 a	28.56 ± 2.97 ab	0.16 ± 0.043 b	1.93 ± 0.26 ab	2.81 ± 0.40 ab
	N_3_P_2_K_1_	62.87 ± 1.98 a	4.23 ± 0.31 ab	0.56 ± 0.14 a	12.25 ± 1.26 a	29.62 ± 1.97 a	0.18 ± 0.033 a	2.14 ± 0.22 a	3.14 ± 0.27 a
	N_4_P_2_K_2_	60.89 ± 0.88 ab	4.33 ± 0.38 ab	0.54 ± 0.055 a	10.88 ± 1.41 a	24.03 ± 1.68 cd	0.17 ± 0.011 b	1.85 ± 0.28 ab	2.49 ± 0.15 bc
Flowering stage (90 d)	CK_3_	87.21 ± 2.01 d	6.40 ± 0.70 b	1.38 ± 0.29 b	24.09 ± 1.78 b	43.81 ± 3.97 c	0.47 ± 0.099 b	5.43 ± 1.03 c	4.47 ± 1.02 b
	N_1_P_1_K_1_	94.49 ± 2.12 c	7.53 ± 1.50 a	1.55 ± 0.32 b	24.52 ± 1.95 ab	45.79 ± 1.44 bc	0.56 ± 0.29 b	7.00 ± 0.52 ab	5.73 ± 0.62 ab
	N_2_P_1_K_2_	101.27 ± 1.51 a	7.09 ± 0.71 b	2.03 ± 1.07 a	26.99 ± 2.10 a	48.14 ± 4.48 bc	0.53 ± 0.13 b	6.51 ± 0.60 bc	5.37 ± 0.57 b
	N_3_P_2_K_1_	97.59 ± 0.77 b	7.01 ± 0.66 b	1.89 ± 0.52 b	25.28 ± 2.35 ab	54.09 ± 3.90 a	0.66 ± 0.19 a	7.44 ± 1.15 ab	6.77 ± 1.64 a
	N_4_P_2_K_2_	98.99 ± 1.92 b	6.93 ± 0.50 b	1.25 ± 0.60 b	24.94 ± 2.52 ab	50.21 ± 4.21 ab	0.47 ± 0.23 b	7.82 ± 1.39 a	5.61 ± 1.14 ab

Note: Values followed by different lowercase letters in the same column indicate significant differences among treatments at the 5% level.

**Table 3 plants-12-04062-t003:** Effect of different NPK ratios on the flower index of ‘Qinhuai Yulian’.

Treatment	Flower Fresh Weight (g)	Flower Dry Weight (g)	Flower Diameter (mm)	Flower Height (mm)	Stem Length (mm)	Maximum Leaf Area (cm^2^)
CK	21.51 ± 0.94 d	2.19 ± 0.32 d	117.58 ± 4.99 d	28.28 ± 1.29 d	82.21 ± 2.01 d	43.21 ± 3.34 d
N_1_P_1_K_1_	25.80 ± 0.89 c	2.81 ± 0.17 c	121.93 ± 1.32 c	32.34 ± 0.70 c	89.49 ± 2.12 c	52.09 ± 2.01 c
N_2_P_1_K_2_	29.95 ± 1.72 b	3.24 ± 0.38 b	126.31 ± 1.90 b	34.47 ± 2.65 b	92.59 ± 0.77 b	56.17 ± 0.90 b
N_3_P_2_K_1_	32.40 ± 1.56 a	3.57 ± 0.24 a	131.73 ± 2.72 a	37.23 ± 1.13 a	98.60 ± 1.30 a	61.10 ± 2.24 a
N_4_P_2_K_2_	28.59 ± 2.07 b	2.93 ± 0.19 bc	121.35 ± 1.71 c	32.43 ± 0.70 c	90.33 ± 0.81 c	52.26 ± 2.75 c

Note: Values followed by different lowercase letters in the same column indicate significant differences among treatments at the 5% level.

**Table 4 plants-12-04062-t004:** Microbial diversity indices of the bacterial and fungal communities in the chrysanthemum rhizosphere soil.

Growth Stage	Treatment	Diversity (Shannon)		Faith’s Phylogenetic Diversity		Richness (Chao1)		Evenness	
		Bacterial	Fungal	Bacterial	Fungal	Bacterial	Fungal	Bacterial	Fungal
Seedling stage (30 d)	CK_1_	8.68 ± 0.10 cdef	5.69 ± 0.08 a	264.15 ± 8.14 abcde	192.99 ± 10.23 ab	3138.56 ± 50.60 cde	1245.51 ± 100.70 abc	0.77 ± 0.01 cdef	0.54 ± 0.01 bcd
	N_1_P_1_K_1_	8.73 ± 0.14 bcde	5.32 ± 0.10 bcd	255.71 ± 9.24 abcdef	197.93 ± 0.94 a	2996.64 ± 109.24 efg	1287.85 ± 60.77 a	0.77 ± 0.01 bc	0.58 ± 0.01 a
	N_2_P_1_K_2_	8.56 ± 0.08 efg	5.24 ± 0.28 bcde	244.27 ± 4.59 def	189.04 ± 8.99 ab	2844.88 ± 25.52 gh	1282.16 ± 125.19 ab	0.76 ± 0.01 cdef	0.53 ± 0.02 cd
	N_3_P_2_K_1_	8.45 ± 0.06 gh	5.02 ± 0.13 efg	234.52 ± 4.26 f	183.73 ± 2.35 abc	2733.81 ± 41.89 h	1188.64 ± 25.12 abcdef	0.76 ± 0.01 ef	0.51 ± 0.01 def
	N_4_P_2_K_2_	8.47 ± 0.10 fgh	4.34 ± 0.12 j	240.06 ± 13.10 ef	162.34 ± 6.37 ef	2755.16 ± 161.19 h	1117.19 ± 104.58 cdef	0.76 ± 0.01 def	0.45 ± 0.01 i
Reproductive stage (60 d)	CK_2_	8.33 ± 0.26 h	5.03 ± 0.10 defg	246.11 ± 13.60 cdef	166.38 ± 4.54 def	2971.85 ± 175.71 fg	1071.91 ± 40.08 ef	0.74 ± 0.02 g	0.53 ± 0.01 de
	N_1_P_1_K_1_	8.92 ± 0.05 ab	4.76 ± 0.22 ghi	252.14 ± 5.24 bcdef	171.37 ± 5.99 cde	3070.27 ± 49.96 def	1147.24 ± 79.45 abcdef	0.79 ± 0.01 a	0.50 ± 0.02 fgh
	N_2_P_1_K_2_	9.03 ± 0.12 a	5.42 ± 0.23 abc	266.75 ± 6.50 abcd	187.46 ± 9.47 ab	3210.32 ± 22.07 bcd	1235.85 ± 50.94 abcd	0.79 ± 0.01 a	0.55 ± 0.02 abc
	N_3_P_2_K_1_	8.92 ± 0.09 ab	5.46 ± 0.16 ab	271.30 ± 0.37 abc	167.44 ± 11.79 def	3278.88 ± 72.26 abc	1108.52 ± 39.02 cdef	0.78 ± 0.01 ab	0.56 ± 0.01 ab
	N_4_P_2_K_2_	8.97 ± 0.11 a	4.52 ± 0.13 ij	279.06 ± 10.00 a	158.97 ± 3.25 ef	3308.27 ± 102.69 ab	1084.61 ± 53.92 def	0.78 ± 0.01 ab	0.47 ± 0.01 hi
Flowering stage (90 d)	CK_3_	8.87 ± 0.02 abc	5.15 ± 0.12 cdef	276.37 ± 8.20 ab	180.96 ± 3.22 bcd	3112.89 ± 114.64 def	1207.80 ± 78.71 abcde	0.77 ± 0.01 bcd	0.53 ± 0.01 cd
	N_1_P_1_K_1_	8.62 ± 0.05 defg	4.82 ± 0.12 gh	272.90 ± 8.36 ab	172.04 ± 5.18 cde	3321.05 ± 23.56 ab	1091.73 ± 57.23 cdef	0.75 ± 0.01 f	0.50 ± 0.01 feg
	N_2_P_1_K_2_	8.96 ± 0.05 a	4.61 ± 0.25 hij	261.17 ± 5.20 abcde	160.18 ± 12.95 ef	3411.55 ± 58.38 a	1073.82 ± 103.85 ef	0.79 ± 0.01 a	0.48 ± 0.02 gh
	N_3_P_2_K_1_	8.87 ± 0.23 abc	4.48 ± 0.16 ij	281.50 ± 3.99 a	153.98 ± 11.14 f	3401.19 ± 22.19 a	1037.86 ± 90.67 f	0.78 ± 0.01 ab	0.47 ± 0.02 hi
	N_4_P_2_K_2_	8.81 ± 0.05 abcd	4.91 ± 0.07 fg	262.30 ± 41.96 abcde	153.86 ± 10.05 f	3330.05 ± 89.09 ab	1127.36 ± 123.04 bcdef	0.77 ± 0.01 cde	0.51 ± 0.01 def

Note: Data are presented as mean ± standard error; different lowercase letters in the same column indicate significant differences among the treatments (*p* < 0.05).

## Data Availability

Data are contained within the article.

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
