# Peer review of "Optimum Nitrogen, Phosphorous, and Potassium Fertilizer Application Increased Chrysanthemum Growth and Quality by Reinforcing the Soil Microbial Community and Nutrient Cycling Function"

_plants, 2023, doi:10.3390/plants12234062_

Round 1

Reviewer 1 Report

Comments and Suggestions for Authors

I appreciate authors to present this work in a good way but to make it more consistant I wana give some commnets and suggestion.

Abstract

Please write percentage of increase of all parameters in abstract results section.

Please remove key words that are already mentioned in the title.

Introduction

Author should write literature study about these nutrients on other crops in few lines .

Materials and methods

Please write longitude and latitude of your experimental site.

Section 2.3: Please remove repeatedly used word "and".

please write PCR protocol in section 2.4.

What is annealing temperature used for Pcr?

Please improve conclusion part also 

Comments on the Quality of English Language

English editing is needed

Author Response

Response to Reviewer 1 Comments

Point 1: Abstract, please write percentage of increase of all parameters in abstract results section.

Response 1: Thank you for your suggestion. Percentage of increased plant growth and plant quality index in abstract were added in the revised manuscript.

Point 2: Abstract, please remove key words that are already mentioned in the title.

Response 2: All the key words that were already mentioned in the title have been revised, please check the revised key words in the revised manuscript.

Point 3: Introduction, Author should write literature study about these nutrients on other crops in few lines.

Response 3: Thank you for your suggestion. We had written literature study about NPK nutrients on other crops in the original manuscript, the species of crops are added and further specified in the revised manuscript.

Point 4: Materials and methods, please write longitude and latitude of your experimental site.

Response 4: Longitude and latitude of experimental site were added in the materials and methods part in the revised manuscript.

Point 5: Section 2.3: Please remove repeatedly used word "and".

Response 5: The title of section 2.3 has been revised to “Soil and plant sampling, laboratory analyses” in the revised manuscript.

Point 6: please write PCR protocol in section 2.4.

Response 6: PCR protocol in section 2.4 was added in the revised manuscript.

Point 7: What is annealing temperature used for Pcr?

Response 7: The annealing temperature used for PCR is 57°C.

Reviewer 2 Report

Comments and Suggestions for Authors

The authors presented an initial experimental design including 5 N levels, 3 P levels and 3 K levels. These gives a combination of 45 treatments. The authors decided to examine only 4 of these 45. Please provide an explanation about that.

There is a lack of information as regards the experimental design and more specifically the soil sampling. It is not obvious about the number of replicates in each treatment and how the rhizosphere soil was collected.

The authors discussed about significant effect in Fig. 2 but there is no statistical tests to support their arguments.

Increase the resolution of the graphs.

Many results are not well presented. A lot of details in description are missing.

The authors presented many analyses concerning the soil microbial communities but a small part of the results of these analyses were discussed in the discussion section. In general the Results section is too large and the Discussion is too short. There must be a balance between them.

Comments on the Quality of English Language

There are some minor linguistic corections that the authors could find together with the other comments in the annotated pdf.

Author Response

Response to Reviewer 2 Comments

Point 1: The authors presented an initial experimental design including 5 N levels, 3 P levels and 3 K levels. These gives a combination of 45 treatments. The authors decided to examine only 4 of these 45. Please provide an explanation about that.

Response 1: Thank you very much for your suggestion. Before this experiment, we carried out a first-round experimental design, which were 5 N levels, 3 P levels and 3 K levels, which in total of 45 treatments indeed, based on the results of preliminary orthogonal test of 45 treatments, 5 representative treatments with significant differences were selected for further study.

Point 2: There is a lack of information as regards the experimental design and more specifically the soil sampling. It is not obvious about the number of replicates in each treatment and how the rhizosphere soil was collected.

Response 2: The information of the experimental design, the number of replicates in each treatment and how the rhizosphere soil was collected were added in the section 2.3 in the revised manuscript.

Point 3: The authors discussed about significant effect in Fig. 2 but there is no statistical tests to support their arguments.

Response 3: Thank you very much for your suggestion. The a, b and c of the Fig. 2 were the statistical analysis not only presented the differences between different NPK treatments on the uptake and tissue distribution of N, P, and K in chrysanthemum plants and also tested the differences in different growth periods of chrysanthemum, they were the main discussion point.

Point 4: Increase the resolution of the graphs.

Response 4: We tried our best to improve the resolution of the graphs, it could be a problem with the magazine template.

Point 5: Many results are not well presented. A lot of details in description are missing.

Response 5: Thank you very much for your suggestion. Our article covered a lot of conclusions, and we had a long discussion part at the time of submission, so, we have to only kept the discussion parts that we were interested in, and which we thought were more important and with significant differences.

Point 6: The authors presented many analyses concerning the soil microbial communities but a small part of the results of these analyses were discussed in the discussion section. In general the Results section is too large and the Discussion is too short. There must be a balance between them.

Response 6: Thank you, there are four paragraphs in the discussion, both the third and the fourth paragraphs were concerning the soil microbial communities.

Reviewer 3 Report

Comments and Suggestions for Authors

1.     The optimal concentrations of N, P, and K had a positive indirect influence on the available soil nutrient content and efficiency of nutrient use by plants by increasing the abundance of Proteobacteria, decreasing the abundance of Actinobacteria, and enhancing the potential functions of nitrogen metabolism pathways. What’s evidence for enhancing the potential function of nitrogen metabolism pathways in this study?

2.     How to change environmental niches of  N, P, and K fertilization combined for C. morifolium cultivation?

3.     Adding a Table of experimental treatment designs in the part of Materials and experimental design;

4.     Figure 2, Figure 4 and is not standardized, please revise and improve it according to the requirements of the journal;

5.     Replacing the legend text in Figure 6 with numbers and explaining the meaning of the replacement numbers in the remarks to increase the readability and aesthetics of Figure 6.

Comments on the Quality of English Language

The overall language of the manuscript is relatively fluent.

Author Response

Response to Reviewer 3’ Comments

Point 1: The optimal concentrations of N, P, and K had a positive indirect influence on the available soil nutrient content and efficiency of nutrient use by plants by increasing the abundance of Proteobacteria, decreasing the abundance of Actinobacteria, and enhancing the potential functions of nitrogen metabolism pathways. What’s evidence for enhancing the potential function of nitrogen metabolism pathways in this study?

Response 1: Thank you very much for your question. As it shows in our study, the optimal concentrations of N, P, and K had a positive indirect influence on the available soil nutrient content and efficiency of nutrient use by plants by increasing the abundance of Proteobacteria, decreasing the abundance of Actinobacteria, and enhancing the potential functions of nitrogen metabolism pathways. N, P, and K fertilization concentrations of 600, 120, and 80 mg·plant-1 were optimal for C. morifolium cultivation, which could change environmental niches and drive the evolution of the soil microbial community and diversity. Shifts in the composition of soil microbes and functional metabolism pathways, such as ABC transporters, nitrogen metabolism, porphyrin, and the metabolism of chlorophyll II, glyoxylate, and dicarboxylate, greatly affected soil nutrient cycling, with potential feedback on C. morifolium nutrient use efficiency and growth. These results provide new insights into the efficient cultivation and management of C. morifolium.

Point 2: How to change environmental niches of  N, P, and K fertilization combined for C. morifolium cultivation?

Response 2: I am sorry, we don’t know ?.

Point 3. Adding a Table of experimental treatment designs in the part of Materials and experimental design;

Response 3: Table of experimental treatment designs in the part of Materials and experimental design (Table 1) was added in the revised manuscript.

Point 4. Figure 2, Figure 4 and is not standardized, please revise and improve it according to the requirements of the journal;

Response 4: Thank you for your suggestion.

Point 5. Replacing the legend text in Figure 6 with numbers and explaining the meaning of the replacement numbers in the remarks to increase the readability and aesthetics of Figure 6.

Response 5: This is a very good suggestion, heatmap is also an acceptable representation of abundant gene-predicted functional profiles in the KEGG pathway of treated rhizosphere soil.

Reviewer 4 Report

Comments and Suggestions for Authors

Manuscript ID: plants-2698904

Optimum nitrogen, phosphorous, and potassium fertilizer application increased chrysanthemum growth and quality by reinforcing the soil microbial community and nutrient cycling function

Yan Xin Fang, Rong Yan Yang, Zhi Guo Zhao et al.

The authors conducted the experiment to investigate the effects of N, P, and K on the growth, nutrient use efficiency, microbial diversity, and composition of Chrysanthemum morifolium. The topic is interesting and relevant, and necessary for growers of these plants in order to improve the volume and quality of flower production. The different rates of NPK fertilizers and their different combinations were studied. One variety (‘Qinhuai Yulian’) of chrysanthemums was chosen as the research object. Will the data obtained in this study be applicable to other varieties of chrysanthemums? What is the opinion of the authors about the reaction of different varieties to fertilization? Can the response be different?

The structure of the article meets the requirements.

Abstract is too long. It must consist of 200 words, according to the requirements for authors.

Materials and methods chapter is detailed, it clearly describes the applied methods. Is the manuscript scientifically sound and the experimental design appropriate to test the hypothesis.

Despite the fact that the methodical part describes the fertilizing options and indicates their abbreviations, I would suggest explaining the meanings of the abbreviations given in the tables below the tables: CK1; N1P1K etc. Specify fertilizer rates.

In subsection 2.3.: Please specify when the soil and plant samples were taken.

Results. The data is discussed in a clear, concise manner. The data are presented with an indication of their statistical significance. The figures and tables are informative and well prepared.

However, I suggest that in figure 1, the numbers on the y-axis be presented in a different format - without specifying the zero after the decimal point (e.g. 20; 40; 60, not 20.00...).

The reference to the pictures in the text is incorrect. According to the requirements, Figure 1, not Fig. 1

References. 26 references (out of 51) are older than 5 years.

The conclusions consistent with the evidence and arguments presented, but I would suggest some corrections in them:

·         In the sentence „Growth indices, flower quality, soil fertility, and nutrient uptake of ‘Qinhuai yulian’ chrysanthemum were significantly promoted using an optimal NPK ratio during growth stages“, I suggest specifying the specific NPK rates that were applied, what you call ‘optimal’.

·         The last sentence of the conclusions is chrestomatous, universally known, it does not say anything new. Either drop it or link it to your research.

After minor revisions, the manuscript is suitable for publication in the MDPI journal Plants.

Author Response

Response to Reviewer 4 Comments

Point 1: Will the data obtained in this study be applicable to other varieties of chrysanthemums? What is the opinion of the authors about the reaction of different varieties to fertilization? Can the response be different?

Response 1: Thank you very much for your question. We had tested the effects of different NPK application rate on 23 different cut Chrysanthemum varieties before, most of them show the similar NPK reqirement, only two varieties show different. We will continue to focus on it in the future.

Point 2: Abstract is too long. It must consist of 200 words, according to the requirements for authors.

Response 2: Thanks, we will further check with the editor.

Point 3: Despite the fact that the methodical part describes the fertilizing options and indicates their abbreviations, I would suggest explaining the meanings of the abbreviations given in the tables below the tables: CK1; N1P1K etc. Specify fertilizer rates.

Response 3: To make it clear, a Table of experimental treatment designs in the part of Materials and experimental design (Table 1) was added in the revised manuscript.

Point 4: In subsection 2.3.: Please specify when the soil and plant samples were taken.

Response 4: Which were specified in the revised manuscript.

Point 5: I suggest that in figure 1, the numbers on the y-axis be presented in a different format - without specifying the zero after the decimal point (e.g. 20; 40; 60, not 20.00...).

Response 5: Thank you very much for your suggestion.

Point 6: The reference to the pictures in the text is incorrect. According to the requirements, Figure 1, not Fig. 1

Response 6: Correction were made in the revised manuscript.

Point 7: The conclusions consistent with the evidence and arguments presented, but I would suggest some corrections in them:  In the sentence, Growth indices, flower quality, soil fertility, and nutrient uptake of ‘Qinhuai yulian’ chrysanthemum were significantly promoted using an optimal NPK ratio during growth stages“, I suggest specifying the specific NPK rates that were applied, what you call ‘optimal’.

Response 7: Correction was made in the revised manuscript.

Point 8: The last sentence of the conclusions is chrestomatous, universally known, it does not say anything new. Either drop it or link it to your research.

 Response 8: Thanks. This sentence is deleted in the revised manuscript.

Round 2

Reviewer 2 Report

Comments and Suggestions for Authors

Dear Editor

 My comments hasn’t been addressed by the authors. There are some general answers but none of them were inserted into the text. The revised version that they provided it’s the same with that with the comments. Probably the authors forwarded the wrong file. Even comments of minor importance hasn’t been addressed. Based on the above I cannot accept the article at its current form.

Point 1: The authors presented an initial experimental design including 5 N levels, 3 P levels and 3 K levels. These gives a combination of 45 treatments. The authors decided to examine only 4 of these 45. Please provide an explanation about that.

Response 1: Thank you very much for your suggestion. Before this experiment, we carried out a first-round experimental design, which were 5 N levels, 3 P levels and 3 K levels, which in total of 45 treatments indeed, based on the results of preliminary orthogonal test of 45 treatments, 5 representative treatments with significant differences were selected for further study.

Revised 2. At least this explanation must be inserted into the text or the authors could say that they tested 4 different combination of N,P and K which were …….., based on a preliminary experiment. 

Point 2: There is a lack of information as regards the experimental design and more specifically the soil sampling. It is not obvious about the number of replicates in each treatment and how the rhizosphere soil was collected.

Response 2: The information of the experimental design, the number of replicates in each treatment and how the rhizosphere soil was collected were added in the section 2.3 in the revised manuscript.

 Point 3: The authors discussed about significant effect in Fig. 2 but there is no statistical tests to support their arguments.

Response 3: Thank you very much for your suggestion. The a, b and c of the Fig. 2 were the statistical analysis not only presented the differences between different NPK treatments on the uptake and tissue distribution of N, P, and K in chrysanthemum plants and also tested the differences in different growth periods of chrysanthemum, they were the main discussion point.

Revised 2. I insist on my question. Are these a,b, c correspond to differences between treatments at the same time points or between times in the same treatment? The authors must explain this in the legend.

Point 4: Increase the resolution of the graphs.

Response 4: We tried our best to improve the resolution of the graphs, it could be a problem with the magazine template.

Point 5: Many results are not well presented. A lot of details in description are missing.

Response 5: Thank you very much for your suggestion. Our article covered a lot of conclusions, and we had a long discussion part at the time of submission, so, we have to only kept the discussion parts that we were interested in, and which we thought were more important and with significant differences.

Point 6: The authors presented many analyses concerning the soil microbial communities but a small part of the results of these analyses were discussed in the discussion section. In general the Results section is too large and the Discussion is too short. There must be a balance between them.

Response 6: Thank you, there are four paragraphs in the discussion, both the third and the fourth paragraphs were concerning the soil microbial communities.

Comments on the Quality of English Language

there is no problem with the quality of the language

Author Response

Response to Reviewer 2th Comments

Point 1: The authors presented an initial experimental design including 5 N levels, 3 P levels and 3 K levels. These gives a combination of 45 treatments. The authors decided to examine only 4 of these 45. Please provide an explanation about that.

Response 1:

Thank you very much for your suggestion again.  I'm really sorry that I may have misunderstood you. I thought I’d just answer the question and resaon, so I didn't insert the answer  into the text. We have insert the sentence “Before this experiment, we have tested 4 different combination of N, P and K which in total of 45 treatments (5 N levels, 3 P levels and 3 K levels), based on the results of preliminary orthogonal test of 45 treatments, five representative treatments with significant differences were selected in this study.” to the revised manuscript. Please check the page3, part 2.2.

Point 2: I insist on my question. Are these a,b, c correspond to differences between treatments at the same time points or between times in the same treatment? The authors must explain this in the legend.

Response 2: The information of the experimental design, the number of replicates in each treatment and how the rhizosphere soil was collected were added in the section 2.3 in the revised manuscript. The explain “A, B, C and D correspond to differences of total nitrogen in root+stem+leaf, total phosphorus and total potassium in root+stem+leaf between treatments at the flowering stage; a, b, c correspond to differences of total nitrogen, total phosphorus and total potassium content between seedling (30d), reproductive (60d) and flowering stages (90d) in the root, stem and leaf resepectively.” was insert in the legend of Fig.2.

Point 4: Increase the resolution of the graphs.

Response 4: Figure 1 was redone to increase clarity and resolution.